# Heavy Water (D_2_O) Containing Preservation Solution Reduces Hepatic Cold Preservation and Reperfusion Injury in an Isolated Perfused Rat Liver (IPRL) Model

**DOI:** 10.3390/jcm8111818

**Published:** 2019-11-01

**Authors:** Shingo Shimada, Moto Fukai, Kengo Shibata, Sodai Sakamoto, Kenji Wakayama, Takahisa Ishikawa, Norio Kawamura, Masato Fujiyoshi, Tsuyoshi Shimamura, Akinobu Taketomi

**Affiliations:** 1Departments of Gastroenterological Surgery I; Hokkaido University Graduate School of Medicine; Kita15-Nishi7, Kita-Ku, Sapporo, Hokkaido 060-8638, Japan; shingoshimada1979@true.ocn.ne.jp (S.S.); kenshiba@pop.med.hokudai.ac.jp (K.S.); soudai114@gmail.com (S.S.); wakayama1977@yahoo.co.jp (K.W.); takahisa0024@yahoo.co.jp (T.I.); fujiyoshi.masato@gmail.com (M.F.); taketomi@med.hokudai.ac.jp (A.T.); 2Transplant Surgery, Hokkaido University Graduate School of Medicine; Kita15-Nishi7, Kita-Ku, Sapporo, Hokkaido 060-8638, Japan; norirv@live.jp; 3Central Clinical Facilities, Division of Organ Transplantation, Hokkaido University Hospital; Kita14-Nishi5, Kita-Ku, Sapporo, Hokkaido 060-8638, Japan; tshima@cocoa.ocn.ne.jp

**Keywords:** heavy water, D_2_O, liver, cold preservation, mitochondria, cytoskeleton

## Abstract

Background: Heavy water (D_2_O) has many biological effects due to the isotope effect of deuterium. We previously reported the efficacy of D_2_O containing solution (Dsol) in the cold preservation of rat hearts. Here, we evaluated whether Dsol reduced hepatic cold preservation and reperfusion injury. Methods: Rat livers were subjected to 48-hour cold storage in University of Wisconsin (UW) solution or Dsol, and subsequently reperfused on an isolated perfused rat liver. Graft function, injury, perfusion kinetics, oxidative stress, and cytoskeletal integrity were assessed. Results: In the UW group, severe ischemia and reperfusion injury (IRI) was shown by histopathology, higher liver enzymes leakage, portal resistance, and apoptotic index, oxygen consumption, less bile production, energy charge, and reduced glutathione (GSH)/oxidized glutathione (GSSG) ratio (versus control). The Dsol group showed that these injuries were significantly ameliorated (versus the UW group). Furthermore, cytoskeletal derangement was progressed in the UW group, as shown by less degradation of α-Fodrin and by the inactivation of the actin depolymerization pathway, whereas these changes were significantly suppressed in the Dsol group. Conclusion: Dsol reduced hepatic IRI after extended cold preservation and subsequent reperfusion. The protection was primarily due to the maintenance of mitochondrial function, cytoskeletal integrity, leading to limiting oxidative stress, apoptosis, and necrosis pathways.

## 1. Introduction

Cold preservation and subsequent reperfusion causes early graft dysfunction [1] and resulting poor outcome in liver transplantation [2]. Cold preservation primarily causes mitochondrial dysfunction and energy depletion [3], initiating oxidative stress [4], Ca^2+^ overload [5], cellular swelling [6], and cytoskeletal disruption [7]. These harmful responses are interrelated and further propagated during the early phase of reperfusion (1–3 hours), and reach apoptosis and/or necrosis thereafter [8]. Accordingly, the significance of interventions at the beginning of reperfusion has been reported [9]. However, it seems necessary to attenuate functional and structural changes during cold preservation, so as not to accelerate dysregulation and injury after reperfusion.

We developed a novel organ preservation solution, named Dsol. Dsol is a modified University of Wisconsin (UW) solution containing 30% D_2_O with high sodium and low potassium concentrations (Table 1) [10]. D_2_O has endemic biological activities, such as the stabilization of microtubules [11], actin cytoskeleton [12], and plasma membrane [13], inhibition of cytosolic Ca^2+^ overload [10], and activation of glucose uptake, glycolysis [14], and mitochondrial respiration [15]. We have reported the efficacy of Dsol in a model of a prolonged cold storage and transplantation of rat hearts, shown by the inhibition of Ca^2+^ overload during cold preservation and by less activation of proteases thereafter [10].

This study was performed in order to evaluate whether Dsol conferred protection against prolonged hepatic cold preservation and reperfusion, and to clarify the mechanism of protection. 

## 2. Materials and Methods 

### 2.1. Chemicals and Reagents

Almost chemicals and reagents were purchased from Wako Pure Chemical Co. Ltd. (Osaka, Japan) unless otherwise noted. Antibodies, phosphorylated-LIM kinase (LIMK)1, Glyceraldehyde 3-phosphate dehydrogenase (GAPDH), and the horseradish peroxidase-conjugated anti-rabbit IgG secondary antibody were purchased from Cell Signaling Technology (Beverly, MA, USA). Phosphorylated-Slingshot was purchased from ECM Biosciences (Versailles, KY, USA). Alpha-Fodrin was purchased from Biomol (Plymouth Meeting, PA, USA).

### 2.2. Preparation of Preservation Solutions

UW solution (Viaspan^®^) was purchased from Bristol-Myers Squibb Co. (New York, NY, USA). Dsol was prepared in our laboratory (Table 1), as previously described [10].

### 2.3. Animals

Male Sprague–Dawley rats (8–9 weeks, 240–280 g) were purchased from Sankyo Labo Service Corporation, Inc (Tokyo, Japan). The feedings and other raising conditions were as previously described [16]. This study was performed by the approval of the Institutional Guide of Hokkaido University for the Care and Use of Laboratory Animals.

### 2.4. Procedures

Rats without fasting were anesthetized by an intraperitoneal injection of pentobarbital sodium (50 mg/kg), and maintained by inhalation of isoflurane, as previously described [9]. Cannulations of bile duct with a PE-10 catheter (Intramedic, Clay Adams, NJ, USA), and abdominal aorta with an 18-G teflon catheter (NIPRO, Osaka, Japan) were performed. The liver was flushed with an ice-chilled Ringer’s lactate solution and 30 mL of University of Wisconsin solution (UW solution) or Dsol after clamp of the descending aorta and incising the proximal end of the clamp. The portal vein was cannulated with a 16-G catheter (NIPRO, Osaka, Japan), and the liver was removed.

### 2.5. Experimental Protocol

Livers were perfused immediately on the isolated perfused rat liver (IPRL) in the control group (CT) (*n* = 6). Cold preservation groups were subdivided into two groups, as follows. In the UW group (UW), livers were preserved in UW for 48 hours (*n* = 6). In the Dsol group (Dsol), livers were preserved in Dsol for 48 hours (*n* = 6). Thereafter, livers were reperfused on the IPRL. 

### 2.6. Isolated Perfused Rat Liver (IPRL) 

The IPRL conditions were set as previously described [9]. At 37 °C and a constant pressure of 8 cmH_2_O in a recirculating system using miniplus3^®^ peristaltic pumps (Gilson Inc., Middleton, WI), livers were perfused with 300 ML of Krebs–Henseleit bicarbonate buffer (KHB), for which pH was adjusted to 7.37–7.43, supplemented with glucose (197 mg/dL) and sodium taurocholate solution (0.3 mM) for 90 min [9]. The perfusate was oxygenated by a silicon tubing and adjusted throughout the experiment (450 < pO_2_ < 550 mmHg).

Portal flow rate was manually measured every 10 min until end of reperfusion. Portal venous pressure (PVP; cmH_2_O) was monitored and automatically recorded. 

Bile was collected every 30 min and expressed as µl/g liver. pO_2_ and pCO_2_ in the perfusate were measured every 30 min using a model 348EX blood gas analyzer (Siemens AG, Munich, Germany).

### 2.7. Sample Collection

Livers at the end of perfusion were weighed and stored at −80 °C until use or fixed in 10% buffered formalin and embedded in paraffin, as described previously [9]. We collected that influent and effluent at 0, 5 min, 30 min, and the end of perfusion (90 min). The activities of aspartate transaminase (AST), alanine transaminase (ALT), and lactate dehydrogenase (LDH) were measured by colorimetric determination with a Hitachi 7020 automatic analyzer (Hitachi, Tokyo, Japan).

### 2.8. IPRL Calculations

Portal venous resistance (PVP), hepatic enzyme released (AST, ALT, LDH), and oxygen consumption rate (OCR) were calculated as follows [9]:PVR [cmH_2_O/mL × min × g liver] = PVP (8 cmH_2_O)/portal flow (mL × min × g liver); 
Hepatic enzyme released (IU/g liver) = [(Ct − C_0_) × V]/LW (g);
where Ct is the enzyme activity (mIU/mL) in the perfusion medium after t min of reperfusion and C_0_ at the end of the stabilization period, V (mL) is the perfusate volume, and LW (g) is the liver weight before preservation. 

OCR (µmol O_2_/min/g liver) = (C_in_ − C_out_) × portal flow (mL/min/g liver), where C_in_ and C_out_ are the O_2_ concentration in the inflow and outflow, respectively. O_2_ concentration (µmol O_2_/mL) = pO_2_ (kPa) × SO_2_ (37 °C) (µmol O_2_/mL/kPa), where SO_2_ (37 °C) is the oxygen solubility in water at 37 °C. SO_2_ (37 °C) = 0.01056 µmol O_2_/mL/kPa [17].

### 2.9. Histology and Apoptotic Index

We stained paraffin-embedded sections with hematoxylin and eosin. Liver injury was assessed by a single pathologist in a blinded manner. Moreover, we stained frozen sections with a fluorescent TdT-mediated dUTP nick end labeling (TUNEL) staining kit (Promega, WI, USA) in accordance with the manufacturer’s instructions, described briefly in the previous report [9]. After mounting the slides with anti-fade reagent Prolong Gold with 4′,6-diamidino-2-phenylindole (DAPI) (Molecular Probes Inc., OR, USA), they were examined by a BZ-9000 fluorescence microscope (Keyence Japan, Osaka, Japan). The apoptotic index was calculated as the number of TUNEL-positive cells, divided by the total number of DAPI-positive cells. Four high power fields (HPFs) were observed per each sample. The average of the four values represents each sample.

### 2.10. ATP and Energy Charge Assay

Tissue adenosine triphosphate (ATP) and energy charge were measured by the high performance liquid chromatography-ultraviolet (HPLC-UV) method [10]. ATP content was expressed as µmol/g liver. Energy charge was calculated as follows:Energy Charge = ([ATP] + 0.5 [ADP])/([ATP] + [ADP] + [AMP]).

### 2.11. Glutathione Assay

Total glutathione (oxidized glutathione (GSSG) + reduced glutathione (GSH)) and oxidized glutathione (GSSG) were measured as described previously [16]. The amounts of total glutathione, GSH, and GSSG were expressed as nano mole GSH equivalent per milligram protein. We determined the redox status of glutathione by the molar ratio of GSH and GSSG (GSH/GSSG).

### 2.12. Western Blot Analysis

Frozen tissue at the end of 48-h cold preservation was homogenized to obtain cytosolic proteins, as previously described [18]. The protein concentration was measured by a BCA Protein Assay Kit (Thermo Scientific, Rockford, IL). Forty micrograms of the proteins was applied to standard sodium dodecyl sulfate (SDS) polyacrylamide gel electrophoresis (SDS-PAGE), transferred onto the poly vinylidene difluoride (PVDF) membrane, and incubated with diluted primary antibodies (1:1000), Alpha-Fodrin, phosphorylated-Slingshot, phosphorylated-LIMK1, and GAPDH, and horseradish peroxidase-conjugated anti-rabbit IgG secondary antibody (1:5000). Protein bands were detected by a chemiluminescent detector Chemi Doc XRS^®^ (Bio-Rad) and were normalized by GAPDH.

### 2.13. Statistical Analysis

Data were expressed as the means ± SD (*n* = 6). We used the Student’s *t*-test or one-way ANOVA in order to evaluate statistical significance, using Stat View 5.0 for Windows (SAS Institute Inc., Cary, NC). A *p*-Value of <0.05 was considered significant. * *p* < 0.05 UW versus Dsol), † *p* < 0.05 versus CT.

## 3. Results

### 3.1. Liver Histopathology and TUNEL Staining

Liver histology at 90-min after reperfusion showed an almost normal appearance in the control group. Severe vacuolization and condensed or swollen nucleus with heterogenous staining were observed in UW group, whereas these changes were mild in the Dsol group (Figure 1). TUNEL staining showed the highest positive cell ratio in the UW group, and less in the CT and Dsol groups (Figure 1).

### 3.2. Liver Function Tests and Apoptotic Cell Ratio

Liver enzymes leakage at 90-min after reperfusion was assessed (Figure 2A,C). The AST, ALT, and LDH activities in the perfusate were 4.60 ± 2.23, 1.37 ± 0.77, and 8.46 ± 4.85 (×10^−1^ IU/g liver) in the control group, respectively. In the UW group, these were 9.18 ± 2.31, 7.91 ± 3.97, and 26.48 ± 4.70 (×10^−1^ IU/g liver), respectively, whereas in Dsol group, ALT and LDH were significantly reduced to 3.60 ± 1.01, and 13.19 ± 12.10 (×10^−1^ IU/g liver), respectively. AST was 6.20 ± 2.53 (×10^−1^ IU/g liver) in the Dsol group (not significant: N.S.). TUNEL-positive cells were almost undetectable in the control group. The TUNEL-positive cell ratio was significantly increased in the UW group, whereas it was significantly decreased in the Dsol group (Figure 2D).

### 3.3. Portal Venous Resistance (PVR)

The graft was perfused at constant pressure (8 cm H_2_O) after 30 min of equilibration perfusion. In the control group, PVR at each time point was maintained at a low value throughout the perfusion, ranging from 3 to 3.5 (cmH_2_O/mL × min × g liver). The UW group showed the value was 6.5 to 7.5, meanwhile, the elevation of PVR was significantly suppressed to around 4.5 in the Dsol group (Figure 3).

### 3.4. Oxygen Consumption Rate (OCR)

Mitochondrial respiratory function was determined by OCR (Figure 4A). At 90-min after reperfusion, OCR in the control group was the highest at 1.42 ± 0.24 (µmol O_2_/min/g liver). It was significantly reduced to 0.67 ± 0.12 (µmol O_2_/min/g liver) in the UW group. Meanwhile, the decrease was significantly suppressed to 1.10 ± 0.16 (µmol O_2_/min/g liver) in the Dsol group. 

### 3.5. Bile Production

Integrated liver function was evaluated by bile production, which was the highest at 98 ± 21 (µL/g) in the control group. It was significantly reduced to 20 ± 5 (µL/g) in the UW group, whereas it was significantly higher in the Dsol group at 58 ± 13 (µL/g) (Figure 4B).

### 3.6. Energy Status

Tissue ATP content at 90-min after reperfusion was the highest at 3.23 ± 0.04 (µmol/g liver) in the control group. It was significantly reduced to 2.11 ± 0.35 and 2.37 ± 0.23 (µmol/g liver) in the UW and Dsol groups, respectively. The ATP of the Dsol group was slightly higher than that of the UW group (*p* = 0.08) (Figure 4C). Energy charges at 90-min after reperfusion were 0.50 ± 0.01, 0.45 ± 0.03, 0.52 ± 0.02, in the control, UW, and Dsol groups, respectively. The energy charge of the Dsol group was slightly higher than that of the UW group (*p* = 0.08) (Figure 4D).

### 3.7. Redox Status of Glutathione

Antioxidant ability and redox status were evaluated. The ratio of reduced and oxidized glutathione (GSH/GSSG; mol/mol) in the control group was 77 ± 5. It was significantly reduced to 39 ± 1 in the UW group, whereas the decrease was significantly suppressed in the Dsol group (52 ± 2) (Figure 4E).

### 3.8. Cytoskeletal Derangement

Calpain and caspase3 related cytoskeletal derangement was evaluated by their substrate (Fodrin) cleavage. Fodrin cleavage was progressed at the end of cold preservation in the UW group, whereas it was significantly suppressed in the Dsol and control groups (Figure 5A). The balance between depolymerization and repolymerization of cytoskeletal actin was assessed by Slingshot and LIMK1. Phosphorylated Slingshot, the inactive form as a Cofilin phosphatase, was the lowest in the UW group, whereas it was higher in the Dsol and control groups (Figure 5B). Phosphorylated LIMK1, the active form of a Cofilin kinase, was the lowest in the UW group, whereas it was maintained at a higher value in the Dsol and control groups (Figure 5C).

## 4. Discussion

We showed the beneficial effects of a novel organ preservation solution, Dsol, against hepatic ischemia reperfusion injury after prolonged cold preservation, as evidenced by well-maintained hepatic functions (bile production, energy status, redox status), reduced tissue injury (liver enzymes leakage, apoptosis), and cytoskeletal integrity (Fodrin cleavage, and the regulation of depolymerization and repolymerization of actin).

IRI was reduced in Dsol group without showing significant changes in ATP content at the end of cold storage (CS) as compared to the UW group (data not shown). Significant differences were observed firstly in PVR within 30-min after reperfusion. Preservation solution containing anti-edema substance prevented reperfusion edema, leading to lower PVR and higher capillary flow, resulting in the amelioration of IRI [19]. Graft weight was reduced to 90% and 96% (versus pre) at the end of preservation in the UW and Dsol groups, and it appeared to be 112.8% and 103.8% (versus pre) after 90-min reperfusion, respectively. It is of note that the graft weight change during 90-min of reperfusion was 22.8% and 8.3% (versus pre) (data not shown). Progressive graft swelling only in UW group may be explained by perfusion with isotonic buffer, while intracellular fluid retained high osmolarity. Accordingly, graft swelling during reperfusion was suppressed in the Dsol group, presumably due to the anti-edema property of D_2_O [13]. Consistent to the report, our observations indicate that anti-swelling property plays an important role in hepatic cold preservation and reperfusion.

OCR is known as a vital marker for mitochondrial function [14]. In this study, OCR, ATP content, and energy charge showed similar results, indicating that better oxygen delivery enabled appropriate oxidative phosphorylation in the Dsol group, thereby showing concomitant reduction of oxidative stress. Mitochondrial intactness was also supported by the amelioration of apoptosis in the Dsol group. Efficacy of D_2_O for the use of organ preservation has been reported in the liver, heart [20], kidney [21], and pancreas [22] by mechanisms involved in the inhibition of the water and Na^+^ influx [15,19], mitochondrial protection, and resulting rapid restoration of ATP content after reperfusion [10]. Among the D_2_O-containing solutions reported, Dsol is the only solution significantly surpassing the ability of UW solution [10]. Here, we confirmed the superiority of Dsol to UW solution in hepatic cold preservation and reperfusion. ATP depletion causes Na^+^/K^+^ ATPase dysfunction, resulting in excessive inflow of Na^+^ [23], which, in turn, causes cytosolic Ca^2+^ overload. Ca^2+^ overload causes mitochondrial permeability transition, the release of cytochrome c, and intrinsic apoptosis thereafter [24]. Consistent with these reports, our data suggested the possible mechanisms of D_2_O mediated graft protection through the maintenance of energy and ions status.

Calpain is known as a Ca^2+^-dependent protease, which is activated with cold and warm ischemic time [25], leading to membrane blebbing via proteolysis of cytoskeletal actin [26]. Alpha-fodrin (spectrin) maintains the function and structure of plasma membrane by connection to the actin skeleton. Thus, degradation of these cytoskeleton-related proteins leads to cellular dysfunction, rupture of bleb, and necrosis [26,27]. Since D_2_O stabilizes cytoskeletal actin [12], tubulin [11], and plasma membrane [13], Dsol is expected to maintain cellular integrity. In fact, Dsol suppressed calpain activity, apoptosis, and necrosis, resulting in the amelioration of graft functions and survival in cold preservation and subsequent heart transplantation. Furthermore, Dsol prevented the elevation of cytosolic Ca^2+^ concentration during cold preservation in vitro [10]. Here, we showed that Dsol prevented cleavage of alpha-fodrin in cold-preserved rat liver.

The actin cytoskeleton plays important roles for the maintenance of structural integrity and cellular functions in normal state [27], while they were disrupted in ischemia and reperfusion, resulting in reduced bile secretion [28]. The cytoskeleton is maintained by the balance between depolymerization and repolymerization of actin. Dysregulation of actin turnover causes a decrease in the F-actin (fiber)/G-actin (monomer) ratio [29]. Dephosphorylated cofilin promotes actin depolymerization by binding to ADP-bound actin, whereas the reaction rate is extremely low with ATP-bound actin due to the change of affinity [30]. Slingshot is active in the dephosphorylated form as a phosphatase, leading to dephosphorylate cofilin, thus promoting actin depolymerization [31]. On the other hand, LIMK1 is active in the phosphorylated form as a kinase, leading to phosphorylate cofilin, thus preventing actin depolymerization [32]. The rapid recovery of ATP content, less cytoskeletal breakdown, and maintenance of actin turnover together with more production of bile in Dsol group are well consistent with these reports [27,28,29,30,31]. Since tissue ATP dynamics are different depending upon the preservation temperature and organ species [33], we speculate that D_2_O acts on the different pathways presumably due to the presence of different proteins in different cells and organs. Further studies are necessary to understand the cell/organ specific and universal effects of protective and harmful biological actions of D_2_O in different temperatures, donor characteristics, and experimental settings, including machine perfusion, isolated reperfusion without blood, and transplantation.

## 5. Conclusions

In conclusion, D_2_O-containing solution, Dsol, conferred protection in rat livers subjected to extended cold preservation and subsequent reperfusion, primarily due to the inhibition of graft edema, thus improving hepatic oxygen delivery and utilization from the early phase of reperfusion. Furthermore, D_2_O prevented cytosolic Ca^2+^ overload and downstream harmful events, including mitochondrial injury, cytoskeletal derangement, and resulting apoptosis and necrosis.

## Figures and Tables

**Figure 1 jcm-08-01818-f001:**
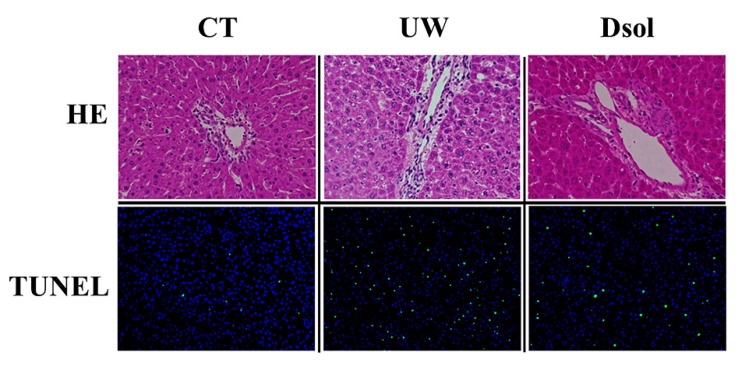
Dsol ameliorates histopathology. H&E staining (20× and 40× magnifications) showed almost normal appearance in the CT group. Vacuolization was shown in the UW group, whereas it was suppressed in the Dsol group. TdT-mediated dUTP nick end labeling (TUNEL)-positive cells were observed in the UW group, whereas they were hardly shown in the CT and Dsol groups.

**Figure 2 jcm-08-01818-f002:**
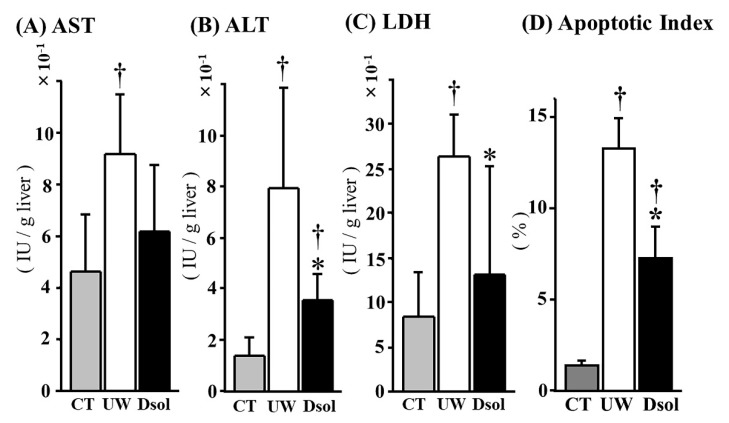
Dsol ameliorates liver enzymes leakage and apoptotic index. Liver enzymes leakage in the perfusate during 90 min of reperfusion was evaluated and expressed as IU/g liver. In the CT group, the AST, ALT, and LDH activities were the lowest. In the UW group, they were significantly augmented, whereas the elevation of ALT and LDH was significantly suppressed in the Dsol group, and AST in the Dsol group was lower than that of UW group, but not statistically significant (**A–C**). Apoptotic index was the lowest in the CT group. In the UW group, it was significantly augmented, whereas the elevation of apoptosis was significantly suppressed in the Dsol group (**D**). Data were expressed as the means ± SD (*n* = 6). *† A *p*-Value less than 0.05 was considered significant: *: UW versus Dsol, † versus CT.

**Figure 3 jcm-08-01818-f003:**
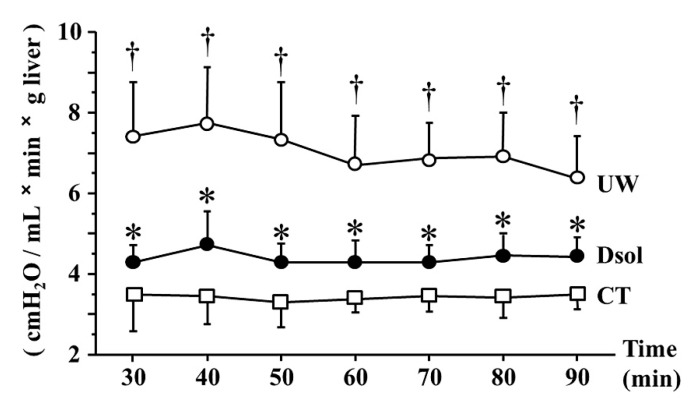
Dsol reduces portal venous resistance (PVR). PVR was the lowest in the CT group (□). It was the highest in the UW group (○), whereas it was significantly reduced in the Dsol group (●) throughout the experiment. Data were expressed as the means ± SD (*n* = 6). *† A *p*-Value less than 0.05 was considered significant: *: UW versus Dsol, † versus CT.

**Figure 4 jcm-08-01818-f004:**
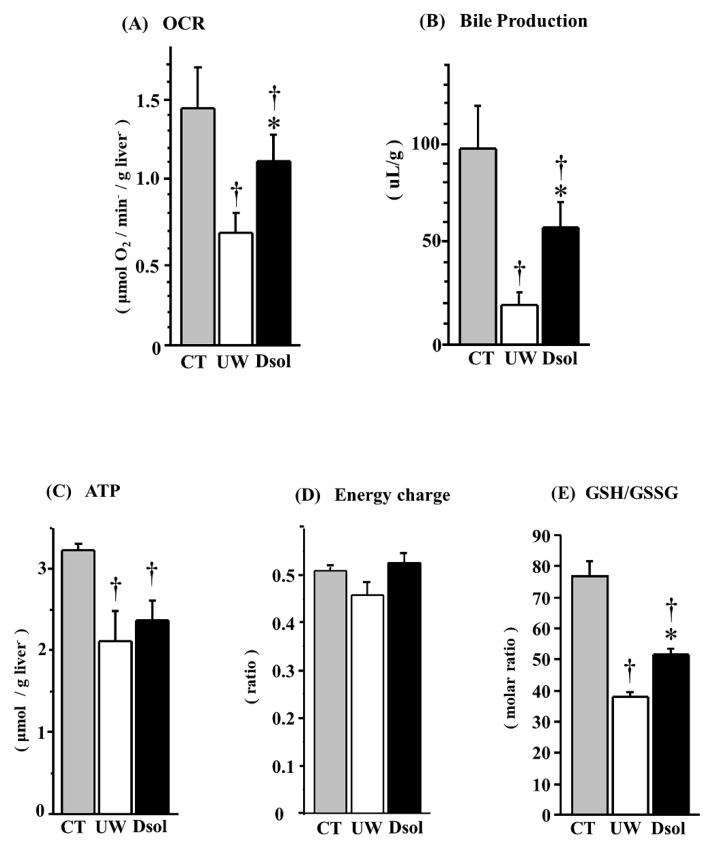
Dsol maintains hepatic functions. Oxygen consumption rate (OCR) at the end of reperfusion and total bile production during 90 min of reperfusion were determined as indices of liver function. In the CT group, OCR (**A**) and bile production (**B**) were the highest. In the UW group, they were the lowest (†), whereas the decreases were significantly suppressed in the Dsol group (*, †). In the CT group, ATP at the end of reperfusion (**C**) was the highest. It was significantly reduced (†) in the both UW and Dsol groups. ATP content in the Dsol group was slightly higher than that of the UW group (statistically not significant). Energy charge at the end of reperfusion was the lowest in the UW group, whereas it was almost equivalent in the CT and Dsol groups (**D**). In the CT group, reduced glutathione (GSH)/oxidized glutathione (GSSG) (**E**) was the highest. In the UW group, it was the lowest (†), whereas the decreases were significantly suppressed in the Dsol group (*, †). Data were expressed as the means ± SD (*n* = 6). *†A *p*-Value less than 0.05 was considered significant: *: UW versus Dsol, †: versus CT.

**Figure 5 jcm-08-01818-f005:**
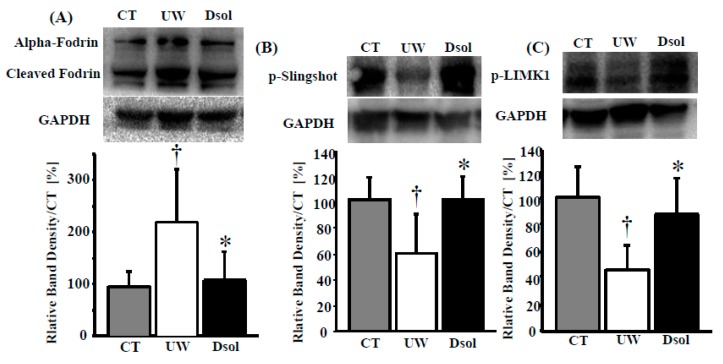
Dsol attenuates cytoskeletal breakdown. Cytosolic protein in the liver was applied to the western blot (*top*), and the relative intensity (*bottom*) was shown. Panels **A–C** show the results for (**A**) Cleaved alpha-Fodrin, (**B**) phosphorylated Slingshot, (**C**) phosphorylated LIMK1. Relative quantitation of each sample was performed, using Glyceraldehyde 3-phosphate dehydrogenase (GAPDH) as an internal control. Each normalized value was further normalized by the mean value in the CT group, and expressed as the means. *†A *p-*value less than 0.05 was considered significant: *: UW versus Dsol, † versus CT.

**Table 1 jcm-08-01818-t001:** Components of Dsol.

Components	Dose
Additives (mM)	
NaOH	125
MgSO_4_	5
KH_2_PO_4_	25
Lactobionate	100
Sucrose	20
Mannitol	10
Adenosine	5
Allopurinol	1
Glutathione	3
Solvent (%)	
H_2_O	70
D_2_O	30
Freezing point (℃)	0.3

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
