# Peer review of "Heavy Water (D2O) Containing Preservation Solution Reduces Hepatic Cold Preservation and Reperfusion Injury in an Isolated Perfused Rat Liver (IPRL) Model"

_jcm, 2019, doi:10.3390/jcm8111818_

Round 1
Reviewer 1 Report
Comments for Authors:
1. on line 6 the first name should be aligned with the others on lines 7 and 8,
2. the image quality that refers to table 1 must be improved and the columns must be better aligned,
3. on line 66 the extra space between “(8-9week,” and “240-280g)” must be removed,
4. on line 94 the extra space between “0,” and “5 min” must be removed and, on the same line, a preposition “at” between “30 min, and” and “the end of” must be added,
5. on line 100 a space between “(8 cmH2O)” and “/” must be added,
6. on line 134 the name of Bio-Rad instrument must be corrected: it’s Chemi Doc, an addictional “i” must be added,
7. on line 144 a space between “TUNEL” and “staining” must be added,
8. The quality of figure 1 must be improved, moreover the dimensions of the panels must be uniformed,
9. On line 147 a space between “histopathology” and “H&E” must be added and the dimension of “x” in 20x magnification must be corrected,
10. I think that figure 2 and the related caption should contain only the graphs related to AST, ALT and LDH, while apoptotic index should be moved to section 3.3. Or, alternatively, you could gather paragraphs 3.2 and 3.3, so in this way you can keep the figure in its current form. Moreover you must add the title of vertical axis of figure 2D,
11. on line 174 the verb “were” must be replaced with “was”,
12. on line 175 spaces must be added in the brackets in this way: (cmH2O / mL x min x g liver), to conform with others in the text,
13. on line 185 the square brackets must be removed,
14. I think you could move figure 4 to the end of the description of each result, so after paragraph 3.8, because as it is now it's difficult to understand. Furthermore the titles of the axes of figures 4D and 4E are missing,
15. on line 198 the word “It.” must be removed,
16. on line 200 a space between “*†” and “ A p value” must be added,
17. on line 203 you must uniform the brackets to the others: numbers outside and units of measurements inside,
18. on line 207 the square brackets must be removed,
19. the quality of figure 5 must be improved and the dimensions of immunoreactive panels must be uniformed, moreover there are thickness differences in the chart axes that must be uniformed,
20. on line 231 a space between “*†” and “ A p value” must be added,
21. on line 244 the extra space between “103.8” and “%” must be removed,
22. on line 267 the word “spectrin” should be placed into brackets and the comma before “spectrin” should be removed,
23. on line 336: reference 15 is not correct, it is the last part of reference 14,
24. on line 351: reference 22 is not correct, it is the last part of reference 21,
25. on line 365: reference 29 is not correct, it is the last part of reference 28.
Author Response
Comments from Reviewer 1
Reviewer: on line 6 the first name should be aligned with the others on lines 7 and 8. Response: Thank you for kind comments. We aligned the first name. Reviewer: the image quality that refers to table 1 must be improved and the columns must be better aligned. Response: Thank you for kind comments. We corrected table 1. Reviewer: on line 66 the extra space between “(8-9week,” and “240-280g)” must be removed.
Response: Thank you for kind comments. We removed the extra space. Reviewer: on line 94 the extra space between “0,” and “5 min” must be removed and, on the same line, a preposition “at” between “30 min, and” and “the end of” must be added. Response: Thank you for kind comments. We revised the followings. “at 0,5,30 min, and at the end of perfusion (90 min)”. Reviewer: on line 100 a space between “(8 cmH2O)” and “/” must be added. Response: Thank you for kind comments. We added a space between “(8 cmH2O)” and “/” Reviewer: on line 134 the name of Bio-Rad instrument must be corrected: it’s Chemi Doc, an additional “i” must be added. Response: Thank you for kind comments. We corrected as “Chemi Doc”. Reviewer: on line 144 a space between “TUNEL” and “staining” must be added. Response: Thank you for kind comments. We added a space between “TUNEL” and “staining”. Reviewer: The quality of figure 1 must be improved, moreover the dimensions of the panels must be uniformed. Response: Thank you for kind comments. We corrected figure 1. Reviewer: On line 147 a space between “histopathology” and “H&E” must be added and the dimension of “x” in 20x magnification must be corrected. Response: Thank you for kind comments. We added a space between “histopathology” and “H&E” and corrected as “20x”. Reviewer: I think that figure 2 and the related caption should contain only the graphs related to AST, ALT and LDH, while apoptotic index should be moved to section 3.3. Or, alternatively, you could gather paragraphs 3.2 and 3.3, so in this way you can keep the figure in its current form. Moreover you must add the title of vertical axis of figure 2D. Response: Thank you for kind comments. We gathered paragraphs 3.2 and 3.3, and added “[%] vertical axis of figure 2D. Reviewer: on line 174 the verb “were” must be replaced with “was”.
Response: Thank you for kind comments. We replaced “were” with “was”. Reviewer: on line 175 spaces must be added in the brackets in this way: (cmH2O / mL x min x g liver), to conform with others in the text. Response: Thank you for kind comments. We corrected as “cmH2O / mL x min x g liver“. Reviewer: on line 185 the square brackets must be removed. Response: Thank you for kind comments. We removed the square brackets. Reviewer: I think you could move figure 4 to the end of the description of each result, so after paragraph 3.8, because as it is now it's difficult to understand. Furthermore the titles of the axes of figures 4D and 4E are missing.
Response: Thank you for kind comments. We added the titles of the axes of figures 4D and 4E. We asked to the editor that figure 4 should be located after paragraph 3.8. Reviewer: on line 198 the word “It.” must be removed.
Response: Thank you for kind comments. We removed “It”. Reviewer: on line 200 a space between “*†” and “ A p value” must be added. Response: Thank you for kind comments. We revised as “† A p value”. Reviewer: on line 203 you must uniform the brackets to the others: numbers outside and units of measurements inside. Response: Thank you for kind comments. We revised as “98±21 (μL/g)”. Reviewer: on line 207 the square brackets must be removed.
Response: Thank you for kind comments. We removed the square brackets. Reviewer: the quality of figure 5 must be improved and the dimensions of immunoreactive panels must be uniformed, moreover there are thickness differences in the chart axes that must be uniformed. Response: Thank you for kind comments. We corrected the figure 5. Reviewer: on line 231 a space between “*†” and “ A p value” must be added.
Response: Thank you for kind comments. We revised as “† A p value”. Reviewer: on line 244 the extra space between “103.8” and “%” must be removed.
Response: Thank you for kind comments. We revised as “8%”. Reviewer: on line 267 the word “spectrin” should be placed into brackets and the comma before “spectrin” should be removed.
Response: Thank you for kind comments. We revised as “Alpha-fodrin (spectrin)”. Reviewer: on line 336: reference 15 is not correct, it is the last part of reference 14.
Response: Thank you for kind comments. We corrected reference 14. Reviewer: on line 351: reference 22 is not correct, it is the last part of reference 21. Response: Thank you for kind comments. We corrected reference 21. Reviewer: on line 365: reference 29 is not correct, it is the last part of reference 28. Response: Thank you for kind comments. We corrected reference 28.

Reviewer 2 Report
This is an experimental model of the use of Heavy Water (D2O) after cold preservation to reduce IRI in livers.
The topic is of interest. Do the authors think that Heavy Water would act differently according to the type of organ or the temperature of preservation? Liver cells are heavily involved with metabolism and ATP production, but would this model apply also for other abdominal organs as for example kidney and pancreas? I would suggest to consider in the discussion also the following: Bellini, M.I.; Yiu, J.; Nozdrin, M.; Papalois, V. The Effect of Preservation Temperature on Liver, Kidney, and Pancreas Tissue ATP in Animal and Preclinical Human Models. J. Clin. Med. 2019, 8, 1421.
In the methods, there is often referral to #9, but some more information in the text would help the reader follow better the flow of the manuscript.
Author Response
Comments from Reviewer 2
Reviewer: Do the authors think that Heavy Water would act differently according to the type of organ or the temperature of preservation? Liver cells are heavily involved with metabolism and ATP production, but would this model apply also for other abdominal organs as for example kidney and pancreas? I would suggest to consider in the discussion also the following: Bellini, M.I.; Yiu, J.; Nozdrin, M.; Papalois, V. The Effect of Preservation Temperature on Liver, Kidney, and Pancreas Tissue ATP in Animal and Preclinical Human Models. Clin. Med. 2019, 8, 1421.
Response: Thank you for very useful comments. Bellini reported that tissue ATP dynamics was different depend on preservation temperature. In addition, the relations between ATP dynamics and temperature might be different by organ species. We agree to the ideas described by Bellini et al.
We confirmed the overall protective effect of Dsol in heart and liver subjected to cold preservation. Although these studies revealed Dsol-mediated protection with similar changes in some indices, whether the dominant acting point is identical or not remains elusive. Since protein expression profiles are different in cardiomyocytes and hepatocytes, the extent of allosteric inhibition by temperature would be different. Accordingly, the influence of D2O (Dsol) may be different. Secondly, net effect of conformational fixation by hydration with H2O/D2O/HDO should be different due to the difference in protein content. Thirdly, the requirement of proton transfer and proton hopping at the surface of water molecules (Grotthuss mechanism) are different in proteins. Therefore, the inhibition rate of these reactions by D2O may be different.
From these considerations, we added the following sentences in Discussion section with a new reference kindly suggested.
“Since tissue ATP dynamics is different depending upon the preservation temperature and organ species [33], we speculate that D2O acts on the different pathways presumably due to the presence of different proteins in different cells and organs. Further studies are necessary to understand the cell/organ specific and universal effects of protective and harmful biological actions of D2O in different temperature, donor characteristics, and experimental settings including machine perfusion, isolated reperfusion without blood, and transplantation.”
33) Bellini MI.; Yiu J.; Nozdrin M.; Papalois V. The effect of preservation temperature on liver, kidney, and pancreas tissue ATP in animal and preclinical human models. J. Clin. Med. 2019, 8, 1421.”.
2.Reviewer: In the methods, there is often referral to #9, but some more information in the text would help the reader follow better the flow of the manuscript.
Response: Thank you for kind comments. We wrote Methods precisely in the first version as reader-friendly and for reproduction. But, plagiarism checker revealed high value (31%). Most of the “self-citation or plagiarism” are in the Methods section, and none in Discussion. Since the experimental setting are completely identical as previously described (#9). It seems difficult to present as different ways. So, we deleted many methods by citing #9.
We added and revised “2.4., 2.5., 2.6., and 2.7.” If resulting revised manuscript shows high value of “plagiarism” beyond the permissive range, we’ll try to erase similarity again according to the editor’s suggestion.
